# Landsat-Derived Annual Maps of Agricultural Greenhouse in Shandong Province, China from 1989 to 2018

Cong Ou [1,2,3] , Jianyu Yang [1,3,]*, Zhenrong Du [1,3,4] , Tingting Zhang [1,3], Bowen Niu [1,3], Quanlong Feng [1,3], Yiming Liu [1,3,5] and Dehai Zhu [1,3]

1   College of Land Science and Technology, China Agricultural University, Beijing 100083, China; oucong@cau.edu.cn (C.O.); duzhenrong@mail.tsinghua.edu.cn (Z.D.); b20203210947@cau.edu.cn (T.Z.); BS20213211013@cau.edu.cn (B.N.); fengql@cau.edu.cn (Q.F.); liuym0086@cau.edu.cn (Y.L.); zhudehai@cau.edu.cn (D.Z.)
2   Institute of Geographic Sciences and Natural Resources Research, Chinese Academy of Sciences, Beijing 100101, China
3   Key Laboratory for Agricultural Land Quality Monitoring and Control, Ministry of Natural Resources, Beijing 100083, China
4   Ministry of Education Key Laboratory for Earth System Modeling, Department of Earth System Science, Institute for Global Change Studies, Tsinghua University, Beijing 100084, China
5   Center of Product Research and Development, China Mobile Communication Group Guangdong Co., Ltd., Guangzhou 510623, China
*   Correspondence: ycjyyang@cau.edu.cn

**Abstract:** Agricultural greenhouse (AG), one of the fastest-growing technology-based approaches worldwide in terms of controlling the environmental conditions of crops, plays an essential role in food production, resource conservation and the rural economy, but has also caused environmental and socio-economic problems due to policy promotion and market demand. Therefore, long-term monitoring of AG is of utmost importance for the sustainable management of protected agriculture, and previous efforts have verified the effectiveness of remote sensing-based techniques for mono-temporal AG mapping in a relatively small area. However, currently, a continuous annual AG remote sensing-based dataset at large-scale is generally unavailable. In this study, an annual AG mapping method oriented to the provincial area and long-term period was developed to produce the first Landsat-derived annual AG dataset in Shandong province, China from 1989 to 2018 on the Google Earth Engine (GEE) platform. The mapping window for each year was selected based on the vegetation growth and the phenological information, which was critical in distinguishing AG from other misclassified categories. Classification for each year was carried out initially based on the random forest classifier after the feature optimization. A temporal consistency correction algorithm based on classification probability was then proposed to the classified AG maps for further improvement. Finally, the average User's Accuracy, Producer's Accuracy and F1-score of AG based on visually-interpreted samples over 30 years reached 96.56%, 86.64% and 0.911, respectively. Furthermore, we also found that the ranked features via calculating the importance of each tested feature resulted in the highest accuracy and the strongest stability in the initial classification stage, and the proposed temporal consistency correction algorithm improved the final products by approximately five percent on average. In general, the resultant AG sequence dataset from our study has revealed the expansion of this typical object of "Human–Nature" interaction in agriculture and has a potential application in use of greenhouse-related technology and the scientific planning of protected agriculture.

**Keywords:** agricultural greenhouse; annual mapping; Landsat; Google Earth Engine

## 1. Introduction

With a growing global population, further urbanization and increasing demand for a balanced food supply, protected agriculture has been widely developed all over the

world, especially in China [1]. Agricultural greenhouses (AG), as the most typical object of protected agriculture, has been steadily increased throughout the world and reached about at a total area of $3.02 \times 10^6$ ha in 2016 [2]. Since it has played a great role in the balanced annual supply of food, the utilization of agricultural resources, the increase of farmers' income and the employment of rural labor, the Chinese government has been vigorously promoting the construction of AG since the late 1980s. China also has the largest area of AG in the world, with a total area of about $1.32 \times 10^6$ ha as of 2016 [3]. In the meantime, the widespread use of greenhouse-related technology also poses certain threats to ecological and environmental safety, the efficient use of cultivated land and national food security. On the one hand, driven by market demand, with the increase in the number of years of continuous cultivation, especially in terms of irrational fertilizer and water management [4] and plastic waste [5] has resulted in continuous crops [6] and soil biodiversity degradation [7]. On the other, the lack of scientific and unified planning for AG construction [8], especially in its early stage of development, largely caused by farmers' autonomous behavior [9], has resulted in the scientific design of the structures not being in accordance with the local geographic location and environment. In addition, as one of the typical ways of non-grain cultivation of cultivated land, greenhouse planting has the potential of disorderly expansion under the will of rational economic man, which reduces the possibility of restoring grain planting [10] and threatens China's most stringent cropland protection policy. In order to prevent and control the surface source pollution caused by the disorderly development of AG, optimize the layout of land for protected agriculture, balance the relationship between non-grain cultivation, ensuring food security, and promoting the sustainable development of agricultural resource utilization, it is urgent to track the long-term dynamics of the AG in the typically protected agriculture regions.

Due to the better availability of satellite images in recent years, remote sensing technology that provides a unique vision for unveiling its growth has been verified to be the most feasible approach to obtain the spatiotemporal information of AG [11]. Several prior studies have proposed various approaches for AG mapping; from the perspective of image classification, it is mainly divided into unsupervised classification and supervised classification. For unsupervised approaches, various index-based approaches have been proposed for AG extraction, such as the Greenhouse Detection Index (GDI) [12], Plastic Greenhouse Index (PGI) or Retrogressive Plastic Greenhouse Index (RPGI) [13], Plastic-Mulched Landcover Index (PMLI) [14] and Index Greenhouse Vegetable Land Extraction (VI) [15]. For supervised classification, the Pixel-based (PB) approach is mainly applied to extract the distribution of AG [16], while the Object-based (OBIA) approach is aimed at AG detection [17–19], delineation [20] or even further for greenhouse crop identification [21,22] in a specific protected agricultural area. In addition to the mono-temporal AG mapping studies mentioned, a few studies have focused on the long-term AG dynamics at regional scales. For instance, Picuno et al. analyzed the plasticulture landscapes changes in southern Italy with two-year intervals over 10 years using multitemporal Landsat TM images [23]. Arcidiacono et al. proposed a model to manage crop-shelter spatial development and used it in a highly representative study of the Italian protected cultivation during 1994–1999 [24]. Ou et al. produced seven greenhouses maps in the 1990–2018 period in a typical protected agricultural region of China [25].

However, none of these studies have achieved a continuous AG mapping over a long time at the provincial scale, which is essential for policymakers to have a complete picture of the development of the protected agriculture and to assess its negative impacts from a macro perspective. With the availability of the Landsat archive, a series of annual maps of urban area [26–28], cropland [29,30], forest [31,32], lake [33] and land cover [34–36] has been carried out. The main challenging part of annual long-term AG mapping is as follows: firstly, the AG is easily mixed with bare cropland and plastic-mulched land at provincial scale; secondly, the construction materials and the planting structure of AG is diverse in different regions and periods, which leads to a strong spatiotemporal heterogeneity of its remote sensing characteristics; and thirdly, the image of individual years is affected by the

cloud cover and strip problems, resulting in the inconsistency of annual maps of AG over the long-term.

To address the above issues, an annual remote sensing mapping method of AG oriented to the provincial area and long-term period was proposed in this study. This method integrated the remote sensing characteristics analysis of provincial AG, mapping window selection, classification feature optimization and temporal consistency correction for annual classification results. Based on this method, we developed the provincial AG annual maps in Google Earth Engine (GEE) at 30 m resolution from 1989 to 2018 over Shandong province, China for the first time. The dataset not only provided the first complete description of the annual dynamics of AG over 30 years at the provincial scale, but also has the potential to serve the use of greenhouse-related technology and the scientific planning of protected agriculture.

## 2. Study Area and Data

### 2.1. Study Area

The study area is Shandong province, which is part of the eastern coastal region of China, between $34°22'\sim38°24'$N and $114°47'\sim122°42'$E (Figure 1), with a total area of about 157,900 km$^2$. This area is dominated by three landforms: mountain, plain and hill, and belongs to a warm temperate monsoon climate zone, with a short spring and autumn and long winter and summer, and the annual average temperature is approximately $11\sim14$ °C. The drainage area of Shandong province is about $4.8\times10^4$ km$^2$, and the average river network density is about 0.24 km/km$^2$. Such climate and hydrological conditions favor agricultural production. Therefore, Shandong is not only a major grain-producing province, but also an important economic crop producing province in China, especially in vegetable production. By the end of 2019, Shandong province has ranked first among the vegetable supply provinces in China with an output of 81.92 million tons, accounting for 11.65% of the total vegetable supply in China. In particular, the large-scale specialized greenhouse-based vegetable planting in this area began in the early 1990s and ranks first among the 34 regions for the number of AG, accounting for 15.78% of the AG in China [37], and is the earliest and largest province to develop protected agriculture in China. Therefore, we selected Shandong province as the representative province in China to trace the full change trajectory of AG in this study.

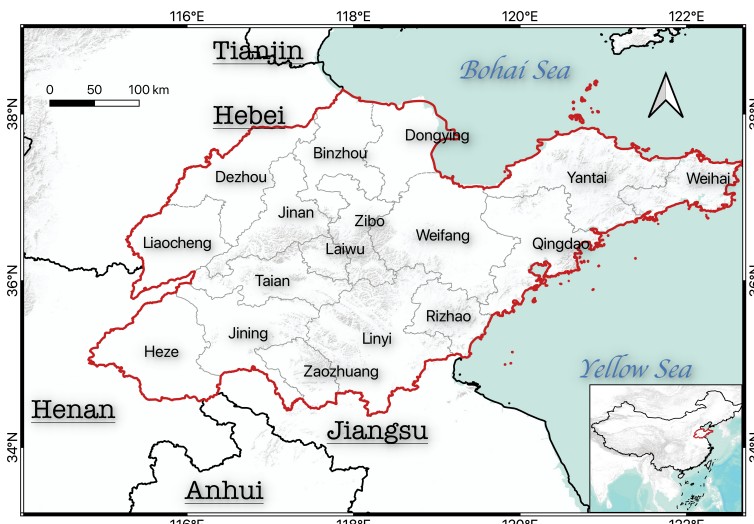

**Figure 1.** Study area.

### 2.2. Landsat Archive

In this study, we used archived Landsat data in its SR (atmospherically corrected surface reflectance) form in GEE from 1989 to 2018 as our remote sensing data, including the Landsat 5 Thematic Mapper (TM), Landsat 7 Enhanced Thematic Mapper Plus (ETM+)

and the Landsat 8 Operational Land Imager (OLI) [38], and the extent of the study area was covered by 16 scenes of Landsat imagery (Figure 2a) for each year. After counting the number of available scenes for each year (Figure 2b), and considering the stripe problem of Landsat 7 ETM + due to the SLC-off issue (when the Scan Line Corrector failed and these products have data gaps) after 2003 [39], we finally selected Landsat 5 images for the period of 1989–2001, 2003–2006 and 2008–2010, Landsat 7 images for period 2002, 2007 and 2011–2012, and Landsat 8 images for the period 2013–2018. In total, around 8450 Landsat scenes were collected for the study area over the past three decades. Since the L1T-level Landsat SR products retrieved in GEE have been corrected for the radiometric, topographic, and atmospheric effects [40], we applied the "CFmask" algorithm to create cloud- and cloud-shadow- free Landsat images covering the study area for each year. In addition, in order to reduce the interference of other similar objects in mountainous and hilly areas, the global DSM data (ALOS World 3D) produced by the ALOS satellite launched by the Japan Aerospace Exploration Agency (JAXA) in 2006 with a spatial resolution of 30 m [41] was used to calculate the elevation and slope of the study area.

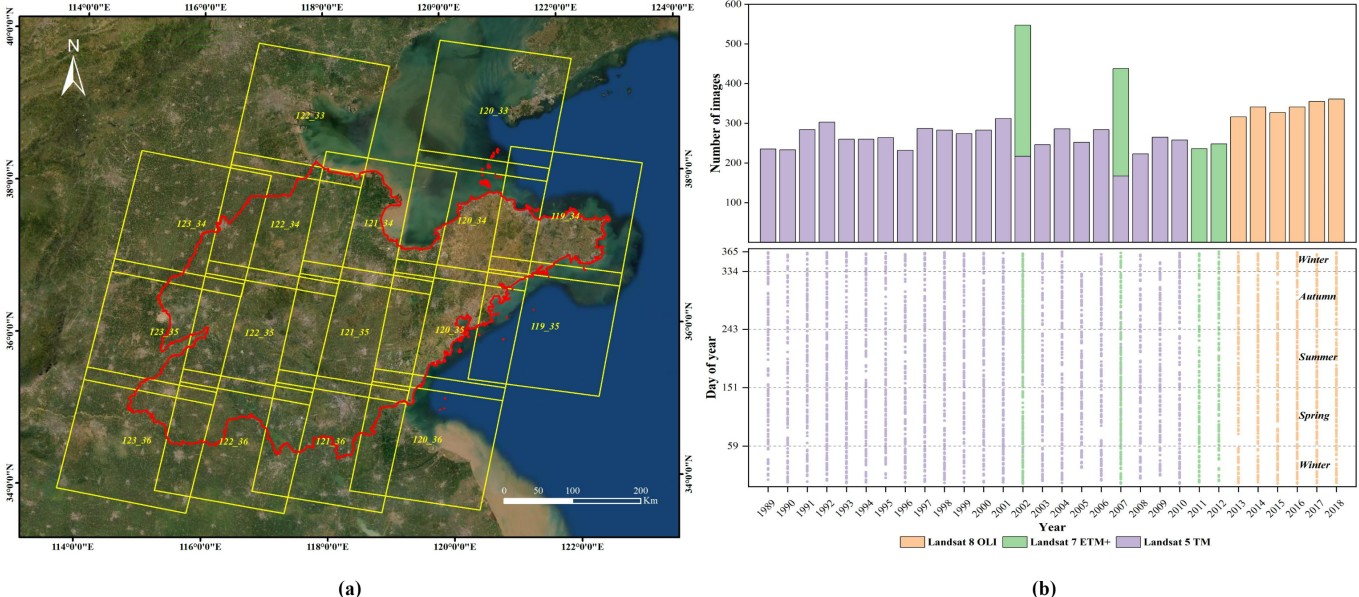

(**a**) (**b**)

**Figure 2.** Spatiotemporal distribution (DOY, day of year) of Landsat images used in this study: (**a**) Landsat tiles in Path 119–123 and Row 33–36, (**b**) the acquisition time of Landsat images.

### 2.3. Reference Dataset for Supervised Training

The annual maps of AG from 1989 to 2018 were derived separately using a supervised machine learning approach in GEE. The accuracy of the reference dataset has a great impact on the classification accuracy under the supervised learning-based strategy, including the accuracy of samples labeling and the spatial distribution of samples [42]. This study is oriented to the provincial area, which covers an area of more than 150,000 km$^2$, while AG in this area are concentrated in several developed regions of protected agriculture. According to the conclusion of the previous study [25], the labeled samples are limited to the above-mentioned regions only, which will lead to large misclassification in other regions. In order to avoid this phenomenon, a 10 km grid sampling structure covering the study area was constructed in this study, which ensured that labeled samples are available in each grid (Figure 3).

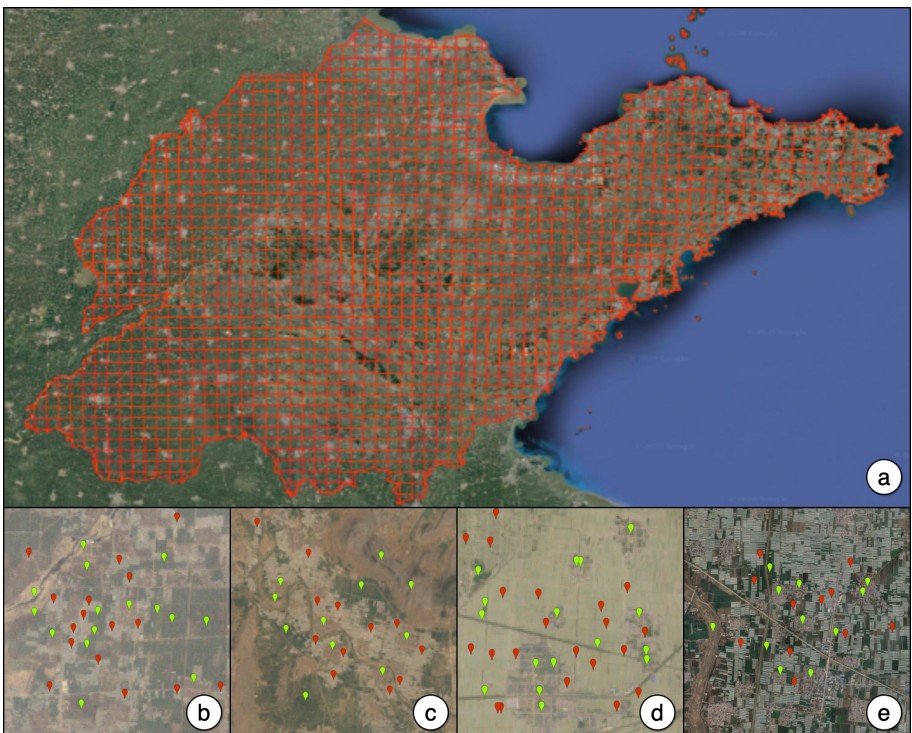

**Figure 3.** (**a**–**e**) The 10 km grid sampling structure: (**a**) and the visual-interpreted samples (red marks for AG, green marks for Non-AG) from Landsat 5 TM (**b**), Landsat 7 ETM+ (**c**), and Landsat 8 OLI (**d**).

Finally, two types of reference datasets, including AG and Non-AG, were labeled via visual inspection of high-resolution imagery such as QuickBird and IKONOS available in Google Earth or Landsat imagery based on the proposed sampling structure. Meanwhile, 70% of this dataset was randomly selected as the training samples and the remaining 30% was randomly selected as the test samples to evaluate the accuracy of classification results for each year. A total of 271,729 samples were finally obtained for the period of 1989–2018, including 44,969 samples for AG and 226,760 samples for Non-AG (Table 1).

**Table 1.** The number of training and testing samples for each class from 1989 to 2018.

| Year | 1989 | 1990 | 1991 | 1992 | 1993 | 1994 | 1995 | 1996 | 1997 | 1998 | 1999 | 2000 | 2001 | 2002 | 2003 |
|---|---|---|---|---|---|---|---|---|---|---|---|---|---|---|---|
| **AG** | 568 | 562 | 612 | 654 | 734 | 756 | 797 | 811 | 937 | 1164 | 1066 | 1181 | 1193 | 1309 | 1512 |
| **Non-AG** | 8252 | 8224 | 8103 | 7890 | 7731 | 7677 | 7631 | 7587 | 7539 | 7536 | 7495 | 7403 | 7367 | 7346 | 7309 |
| **Total** | 8820 | 8786 | 8715 | 8544 | 8465 | 8433 | 8428 | 8398 | 8476 | 8700 | 8561 | 8584 | 8560 | 8655 | 8821 |
| **Year** | **2004** | **2005** | **2006** | **2007** | **2008** | **2009** | **2010** | **2011** | **2012** | **2013** | **2014** | **2015** | **2016** | **2017** | **2018** |
| **AG** | 1776 | 1604 | 1516 | 1941 | 1958 | 1948 | 1890 | 2078 | 2049 | 1849 | 1908 | 2020 | 2099 | 2333 | 2308 |
| **Non-AG** | 7262 | 7159 | 7141 | 7141 | 7117 | 7108 | 7073 | 7066 | 7107 | 7009 | 6900 | 6852 | 6831 | 6745 | 6653 |
| **Total** | 9038 | 8763 | 8657 | 9082 | 9075 | 9056 | 8963 | 9144 | 9156 | 8858 | 8808 | 8872 | 8930 | 9078 | 8961 |

## 3. Methods

This study was aimed at developing a set of continuous annual AG maps at 30 m resolution for Shandong province, China for the period of 1989–2018, and the research workflow mainly included four parts: collecting the Landsat archive and reference dataset, remote sensing characteristics analysis and mapping window selection, input feature optimization and annual classification, as well as temporal consistency correction and accuracy evaluation (Figure 4). The whole procedure was implemented on the GEE platform, which has shown great potential in dealing with massive image processing for multi-temporal classification in previous studies [43–45]. Based on the obtained and

analyzed Landsat archive and reference dataset (Sections 2.2 and 2.3), we (1) analyzed the spectral curves of different landcover categories and its NDVI time series, combining the phenological information of cropland and plastic-mulched land, selected the best mapping window within a year for annual AG mapping, (2) calculated the tested features and obtained the best feature combination via evaluated average importance over 30 years for each feature, and derived the initial classifications, and (3) applied a temporal consistency correction algorithm to generate the final annual AG classifications and performed accuracy evaluation. A detailed description of each individual part of the whole procedure is presented in the following sections.

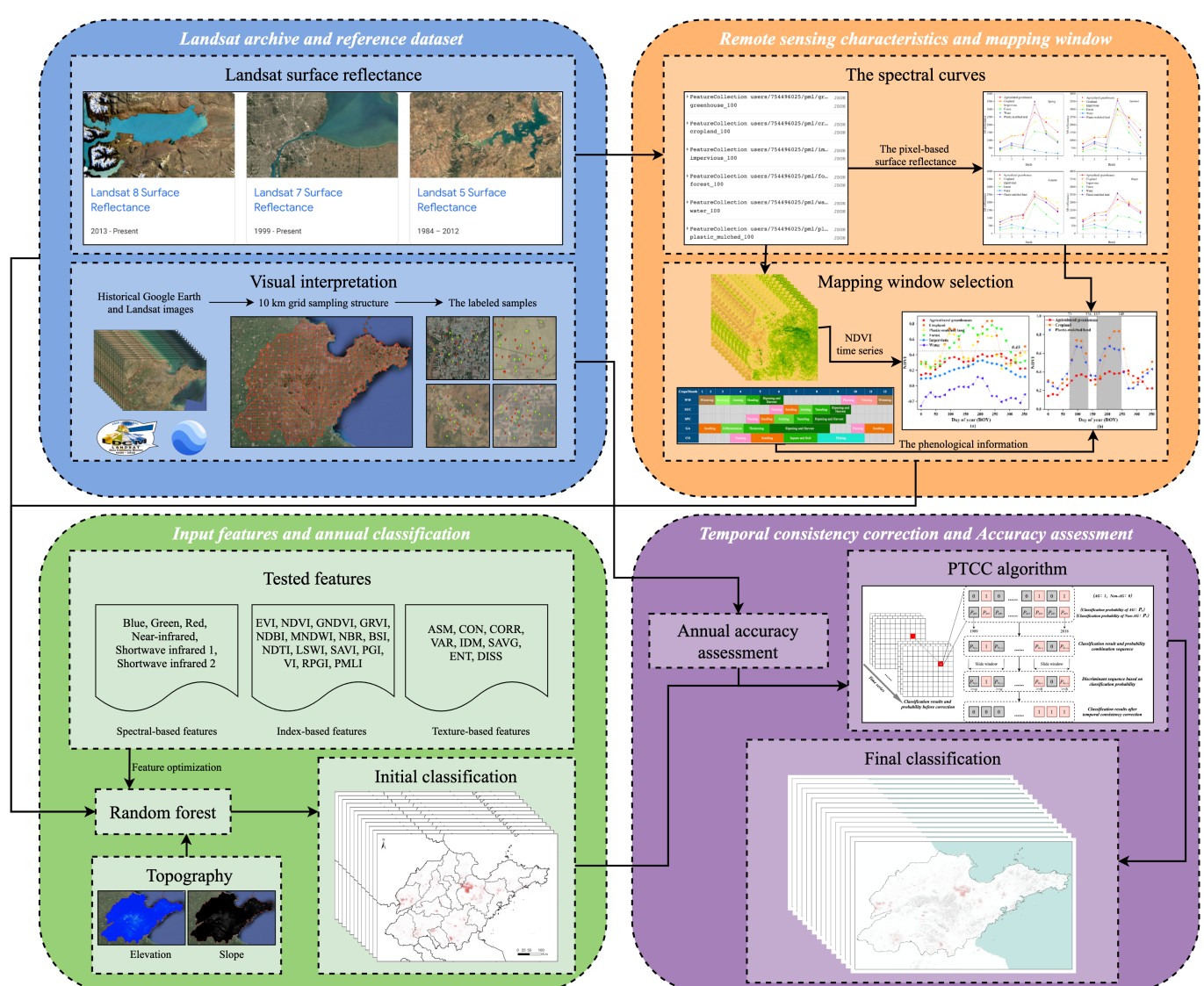

**Figure 4.** The research flowchart of the article.

## 3.1. Remote Sensing Characteristics Analysis

As a kind of agricultural facility to ensure the balanced annual output of crops, the spectral curves of AG will be affected by the change of crop phenology, but theoretically, its seasonal variation is more stable than that of cropland or forest. Therefore, the quantitative analysis of the spectral curves of AG is an important basis for mapping AG. In this study, 600 samples (divided into AG, cropland, impervious, forest, water and plastic-mulched land, 100 samples in each category) were individually labeled on the Landsat 8 OLI imagery in 2018, and the pixel-based surface reflectance of those types was used to analyze its spectral curves in different seasons. In order to reduce the interference

of outliers among them, the median value of each band corresponding to each type was used as the SR reflectance of each band.

As shown in Figure 5, firstly, the spectral curves of AG are more stable than cropland, forest and plastic-mulched land and more variable than impervious and water in different seasons, which indicates that vegetation cover is a determinant of seasonal variations in the spectral curves of all land cover types, and differences in vegetation growth are an important point for differentiating AG from other types. Secondly, the spectral curves of AG are similar to those of cropland and plastic-mulched land that leads to a poor separability in autumn and winter, while the spectral curves of AG are obviously different to those of impervious, water and forest that leads to a strong separability in all seasons, which indicates that those non-crop growing land cover types interfere less with the remote sensing identification of AG. Thirdly, in terms of specific bands, the visible band (2, 3, 4) of AG is close to impervious in spring and summer, and close to cropland and plastic-mulched land in autumn and winter; the infrared band (5) of AG is close to the forest in summer, and close to cropland, plastic-mulched land and impervious in autumn, but differs from all other land cover types in spring and winter; the short-wave infrared band (6 and 7) of AG differs from all other land cover types in spring and summer, which indicates that all the above-mentioned bands can play a role for AG mapping, and the key lies in the selection of the mapping window.

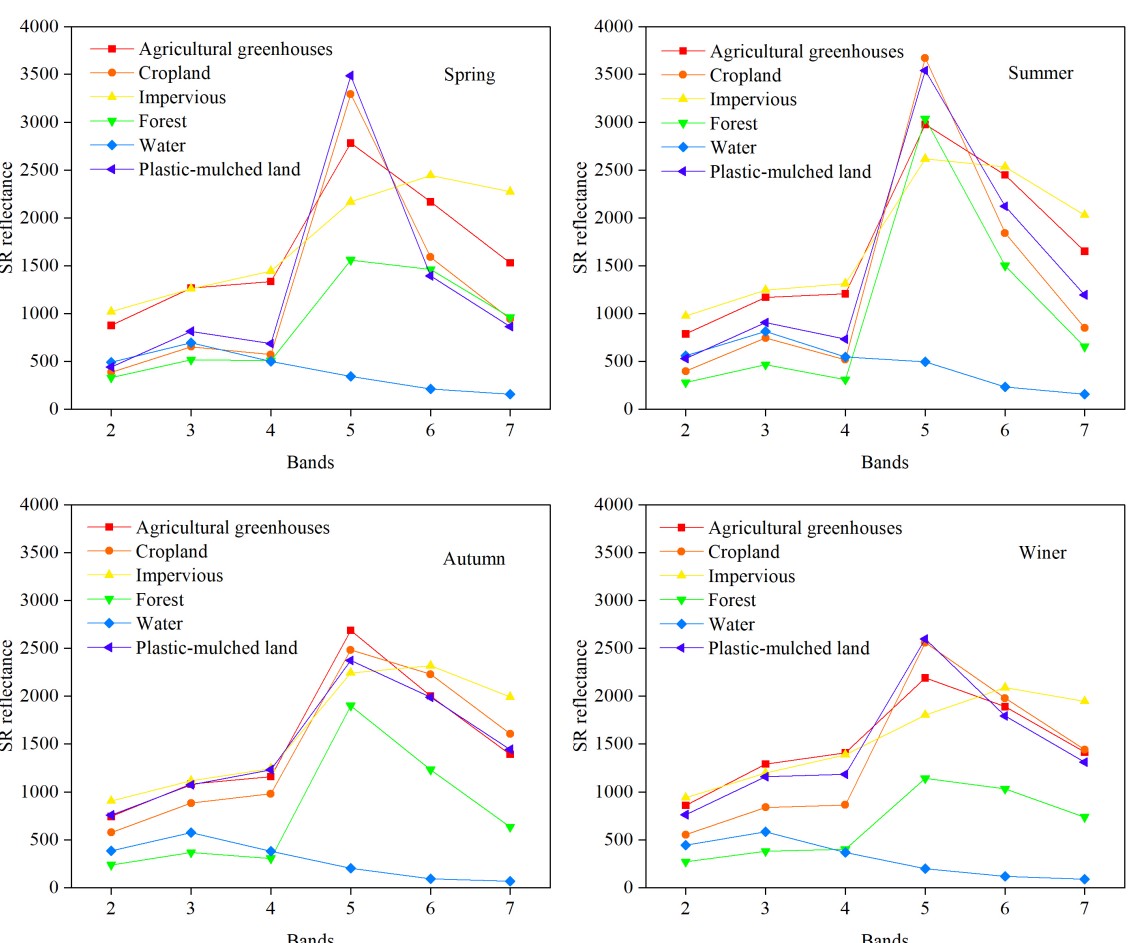

**Figure 5.** Spectral curves of different land cover types (SR reflectance in median value) in different seasons.

### 3.2. Mapping Window Selection

According to the results of the spectral curves analysis, cropland and plastic-mulched land are most likely to cause misclassification in AG mapping and their differences in vegetation growth during a year are critical to distinguishing them. Therefore, the selection

of the mapping window should consider the vegetation growth that best differentiates AG from cropland and plastic-mulched land. Based on this assumption, we constructed the time-series NDVI data of AG, cropland and plastic-mulched land at a 16-day interval from the Landsat 8 OLI imagery in 2018. Specifically, we calculated the median NDVI value of all the samples and used it to represent the NDVI value for each category, and we further smoothed the NDVI time series by the Savitzky-Golay filter to eliminate small fluctuations using a moving window of size 5 and a filter order of 2 [46]. As shown in Figure 6, there are two periods of flourishing vegetation growth within a year, in which the difference of the NDVI value between AG, cropland and plastic-mulched land is pronounced. The first period is day of year (DOY) 73-136 (from 15 March to 15 May), in which the maximum NDVI difference between AG and cropland is about 0.4, and that between AG and plastic-mulched land is about 0.3; The second period is DOY 165-248 (from 15 June to 5 September), in which the maximum difference of NDVI between AG and cropland is about 0.5, and that between AG and plastic-mulched land is about 0.3.

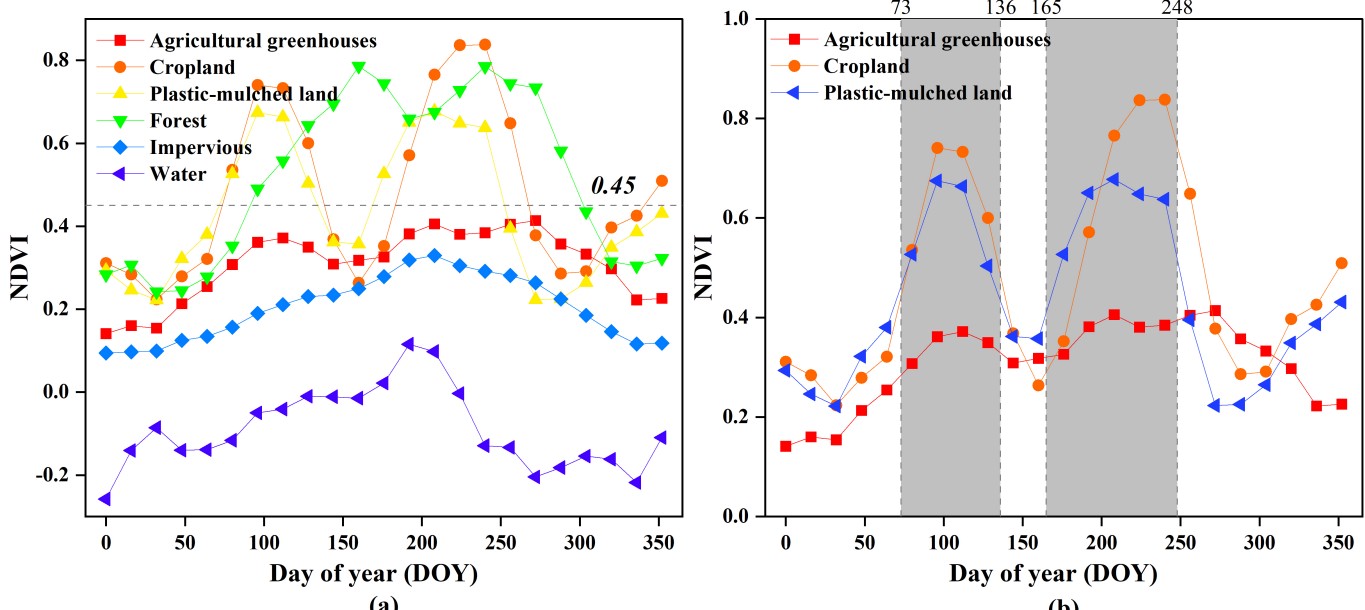

**Figure 6.** (**a**,**b**) Mapping window selection based on vegetation growth.

Due to the complex crop structure in AG, we collected the phenological information of cropland and plastic-mulched land in Shandong province (Figure 7). Winter wheat (WW), summer corn (SUC) and spring corn (SPC) are the dominant crop type of cropland in Shandong province, DOY 73-136 are mainly in the reviving, jointing and heading stage of WW and the planting and seeding stage of SPC, and DOY 165-248 are mainly in the ripening and harvest stage of WW, and the seeding, jointing, tesseling and repending-harvest stage of SUC or SPC. Meanwhile, garlic (GA) and cotton (CO) are the dominant crop type of plastic-mulched land in Shandong province, DOY 73-136 are mainly in the differentiation stage of GA and the planting and seeding stage of CO, and DOY 165-248 are mainly in the repending-harvest stage of GA and the square-boll and picking stage of CO. Referring to the crop calendars for the dominant crops in Shandong province, both mapping windows are for the period that the vegetation growth is flourishing. However, as the major gain production crop in Shandong province, WW is usually harvested in late DOY 165-248, while all kinds of crops are usually grown in DOY 73-136, which shows a better differentiation AG from cropland and plastic-mulched land than DOY 165-248. Therefore, combining the smoothed NDVI times-series and the phenological information, we finally selected DOY 73-136 (from 15 March to 15 May) as the best mapping window within a year for annual AG mapping.

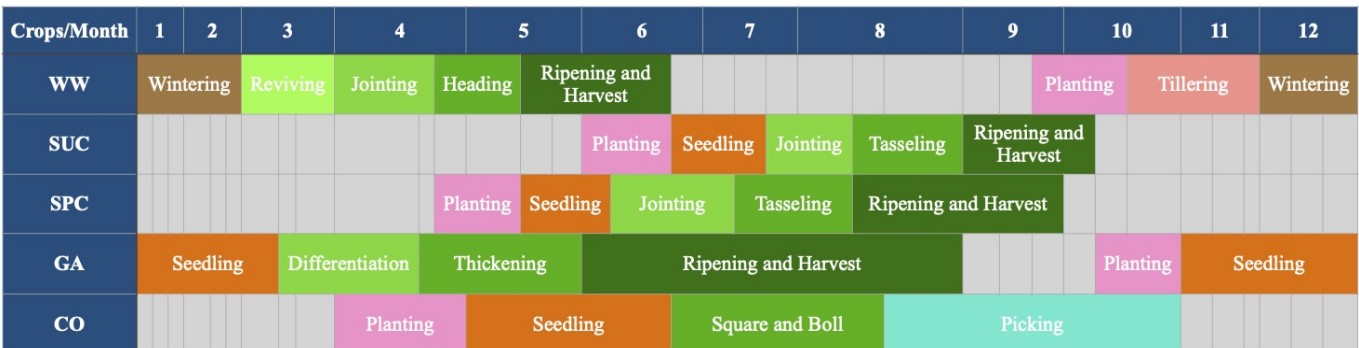

**Figure 7.** The phenological information of cropland and plastic-mulched land in Shandong province includes winter wheat (WW), summer corn (SUC), spring corn (SPC), garlic (GA) and cotton (CO).

### 3.3. Tested Features

The AG exhibits wider feature variation with respect to both space and time due to the difference in construction type, crop structure and satellite sensors. In order to reduce the mapping error in classification accuracy due to the different contributions of features to AG classification in different years and different regions, the tested features were calculated in terms of spectral-based, index-based, and texture-based features (Table 2). Firstly, based on all available Landsat SR within the selected mapping window, we calculated the median value of the image composite for each spectral band [47], including blue (B), green (G), red (R), near-infrared (NIR), shortwave infrared 1 (SWIR1) and shortwave infrared 2 (SWIR2).

Given that the index-based features can effectively enhance the information of spectral-based features in specific directions, we selected 15 index-based features based on the analysis of remote sensing characteristics of greenhouses and previous research on greenhouse mapping. Enhanced Vegetation Index (EVI), The Normalized Difference Vegetation Index (NDVI), Green NDVI (GNDVI) and Green Red Vegetation indices (GRVI) were used to indicate the vegetation greenness, and Normalized Difference Built-up Index (NDBI) was one of the most popular indices for mapping impervious areas. The Modified Normalized Difference Water Index (MNDWI) and Land Surface Water Index (LSWI) were closed to open surface water bodies and land surface moisture, respectively. Normalized Burn Ratio (NBR) was a good indicator of surface temperature, Bare Soil Index (BSI) and Soil Adjusted Vegetation Index (SAVI) were sensitive to the background information from vegetation and soils, and Normalized Difference Tillage Index (NDTI) was used to distinguish the differences of residue between AG, cropland and plastic-film land. Plastic Greenhouse Index (PGI), Index Greenhouse Vegetable Land Extraction (VI), Retrogressive Plastic Greenhouse Index (RPGI) and Plastic-Mulched Landcover Index (PMLI) were both developed to delineate greenhouse or plastic-film by previous studies. It should be noted that NBR, LSWI and NDTI were first used to identify AG.

Since the construction of a greenhouse can significantly change the texture of the land surface, we further calculated texture-based features based on the gray-level co-occurrence matrix (GLCM) [48]. Previous studies have shown that the blue band is the most sensitive to the greenhouse fraction in each Landsat band [13], therefore we selected the blue band for texture-based features calculation. Referring to the results of previous studies [21,22] and the texture characteristics of the greenhouse in the study area (e.g., contrast, correlation, and entropy, etc.), the eight most common texture-based features were selected and computed via the "glcmTexture" function in GEE, including the angular second moment (ASM), contrast (CON), correlation (CORR), variance (VAR), inverse difference moment (IDM), sum average (SAVG), entropy (ENT) and dissimilarity (DISS).

**Table 2.** Landsat time-series features. GLCM, gray-level co-occurrence matrix; NIR, near-infrared; SWIR1, shortwave infrared 1; SWIR2, shortwave infrared 2; R, red band; G, green band; B, blue band.

| | Tested Features | Description | Reference |
|---|---|---|---|
| Spectral-based features | Median value | Median value of B, G, R, NIR, SWIR1, and SWIR2 bands | [47] |
| Index-based features | EVI | 2.5*(NIR-R)/(NIR+6*R-7.5*B+1) | [49] |
| | NDVI | (NIR-R)/(NIR+R) | [50] |
| | GNDVI | (NIR-G)/(NIR+G) | [51] |
| | GRVI | (G-R)/(G+R) | [52] |
| | NDBI | (SWIR1-NIR)/(SWIR1+NIR) | [53] |
| | MNDWI | (G-SWIR1)/(G+SWIR1) | [54] |
| | NBR | (NIR-SWIR2)/(NIR+SWIR2) | [55] |
| | BSI | ((SWIR1+R)-(NIR+B))/((SWIR1+R)+(NIR+B)) | [56] |
| | NDTI | (SWIR1-SWIR2)/(SWIR1+SWIR2) | [57] |
| | LSWI | (NIR-SWIR1)/(NIR+SWIR1) | [58] |
| | SAVI | (1.5*(NIR-R))/(NIR+R+0.5) | [59] |
| | PGI | (100*R*(NIR-R))/(1-(NIR+B+G)/3) | [60] |
| | VI | ((SWIR1-NIR)/(SWIR1+NIR))*((NIR-R)/(NIR+R)) | [11] |
| | RPGI | (100*B)/(1-(NIR+B+G)/3) | [13] |
| | PMLI | (SWIR1-R)/(SWIR1+R) | [61] |
| Texture-based features | ASM | Angular Second Moment of GLCM from B | [48] |
| | CON | Contrast of GLCM from B | [48] |
| | CORR | Correlation of GLCM from B | [48] |
| | VAR | Variance of GLCM from B | [48] |
| | IDM | Inverse Difference Moment of GLCM from B | [48] |
| | SAVG | Sum Average of GLCM from B | [48] |
| | ENT | Entropy of GLCM from B | [48] |
| | DISS | Dissimilarity of GLCM from B | [48] |

*3.4. Classification and Feature Optimization*

As the research flowchart described, we adopted the supervised learning method to produce AG classification for each year. Random forest (RF) classifier, which is a combination of tree predictors such that each tree depends on the values of a random vector sampled independently and with the same distribution for all trees in the forest [62], has a number of advantages, such as the additional description of data, less manual intervention, the ability to handle high-dimensional data, and the robustness to missing data [63–65]. It has also been widely used for large-scale mapping applications [66]. Therefore, the RF classifier was used to generate the annual AG maps in this study. The number of trees ($N$) was set to 100 according to the total number of features and a large number of experiments [28], and the decision tree divides nodes ($M$) was set to six based on the arithmetic square root of the total number of features [67].

In addition, for the classifier based on machine learning algorithm, it is not that the more feature dimensions, the higher the final classification accuracy. The fact that certain redundancy between features and exorbitant feature dimension often causes "dimension disaster", results in the decline of classifier learning ability [68]. During the construction of RF, about 63% of training samples are bootstrapped for the training of the classifier. Meanwhile, about 37% of training samples are not extracted, which is called "Out-of-Bag (OOB)" data. The OOB data not only be used to determine whether the training stage is completed, but also can be used to calculate the importance of each feature. Therefore, we promoted feature optimization by calculating the importance of each tested feature in the process of RF construction.

## 3.5. Temporal Consistency Correction

The selected mapping window within a year for annual AG mapping is Spring, in which the weather of Shandong province is changeable, which leads to different cloud cover phenomenon every year. In addition, the SLC-off issue of Landsat 7 after 31 May 2003 led to the stripe problem of classification results in 2007, 2011 and 2012. Therefore, it is necessary to modify the long-term mapping results to reduce the mapping errors caused by the inconsistent image quality each year. The long-term mapping error could be reduced by checking whether specific land cover change is reasonable under the temporal context [69], and the basic idea of previous research was to eliminate it by using a temporal filtering algorithm based on the temporal context (previous and after classification results in specific sliding window) [36,70].

In this study, a new algorithm based on classification probability (Probability-based Temporal Consistency Correction, PTCC) was proposed to obtain more consistent long-term mapping results than the individual results. Detailed procedures of the PTCC algorithm are shown in Figure 8, the individual classification results and its classification probability of each pixel were obtained, and then the individual classification results and its adjacent classification probability in the specific sliding window of each pixel were combined. For those pixels that were recognized as AG, if its adjacent classification probability were both greater or equal to threshold *a*, it would be corrected to Non-AG. For those pixels that were recognized as Non-AG, if its adjacent classification probability were both greater or equal to threshold *b*, it would be corrected to AG. In light of the demolition and construction characteristics of AG, the sliding window of this study was set to 3 (only the classification probability of the previous year and the next year is considered). Therefore, it can be seen that the key point to implement the PTCC algorithm is to set threshold *a* and *b* when the sliding window is determined. Compared to the related research, the revisability of long-term mapping results with different classification probabilities was discussed in this study.

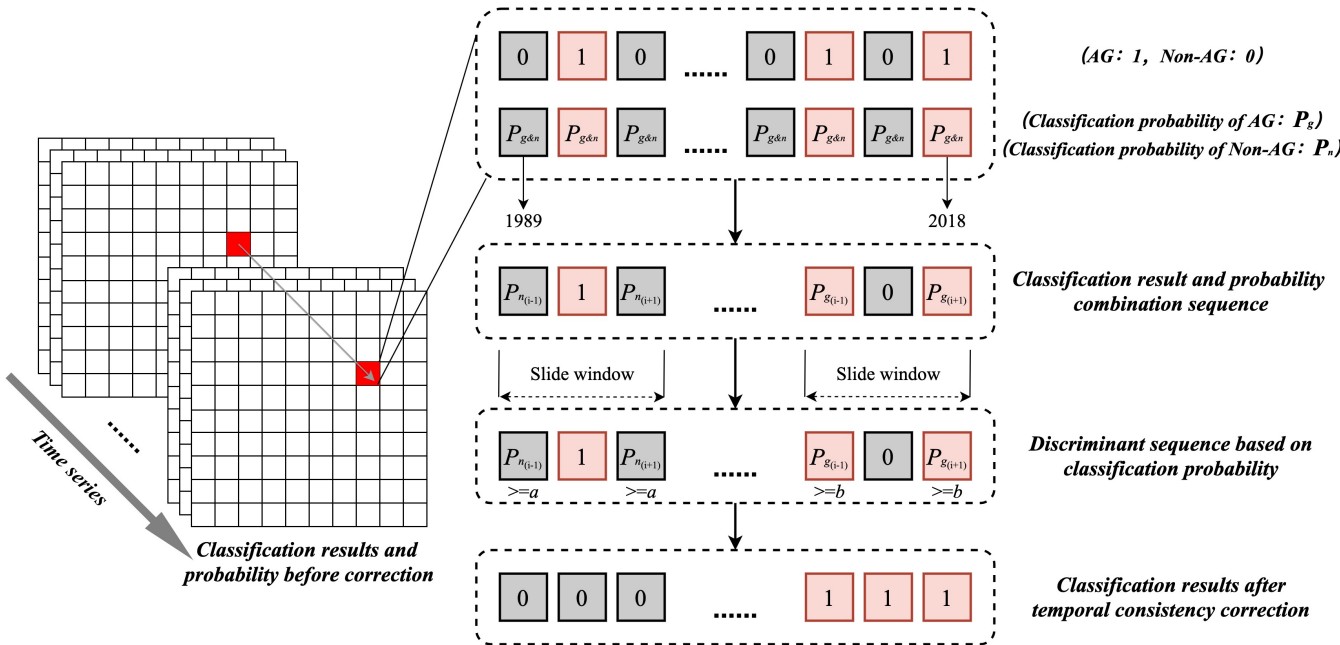

**Figure 8.** Detailed procedures of the PTCC algorithm.

## 3.6. Accuracy Assessment

In order to evaluate the accuracy of the annual classification results under different scenarios for feature optimization and temporal consistency correction, we calculated the annual classification accuracy based on the confusion matrix. Since the classification of AG in this study is a two-class task, the confusion matrix is a 2-by-2 matrix, including

true AG (TAG), false AG (FAG), true non-AG (TNAG) and false non-AG grapes (FNAG). The Producer's Accuracy (PA) and User's Accuracy (UA) for each category are calculated as:

$$UA_i = \frac{TP_i}{P_i} \times 100\% \tag{1}$$

$$PA_i = \frac{TP_i}{C_i} \times 100\% \tag{2}$$

where $TP_i$ represents the total number of pixels correctly classified into a certain category (TAG or TNAG), $P_i$ represents the total number of pixels of a certain category in the whole classification result (TAG+FAG or TNAG+FNAG), and $C_i$ represents the total number of pixels of a certain category in the test samples (TAG+FNAG or TNAG+FAG). Finally, The F1-score conveys the balance between PA and UA, which is suitable for the unbalanced samples of the two-class task, is computed as:

$$F1_i = 2 \times \frac{UA_i \times PA_i}{UA_i + PA_i} \tag{3}$$

## 4. Results and Discussion

### 4.1. Performance of Feature Optimization

In the process of features optimization, we first performed the unbiased estimates of the generalization errors for each feature, and computed the importance of all features from 1989 to 2018. In terms of the different size of OOB data in each year, the importance of each feature was normalized over 30 years, and then the contribution of each feature for AG classification in each year was obtained. As shown in Figure 9a, the importance of spectral features has the smallest variation in 30 years, the importance of index features has the largest variation in 30 years, and the importance of texture features has a slight variation in 30 years. Furtherly, we ranked the average importance of each feature over 30 years. Figure 9b showed that GREEN (0.33) and BLUE (0.28) got the highest average importance among spectral features, NDTI (0.88) and PGI (0.21) got the highest average importance among index features, and SAVG (0.41) and CON (0.35) got the highest average importance among texture features. Notably, we found that NDTI (0.88) and NBR (0.46), which were first applied to remote sensing recognition of AG, ranked first and seventh in the average importance respectively. It means that in addition to the growth of vegetation in AG, the thermal characteristics of AG can also be used as an important factor in AG classification. Finally, all features were divided into four combinations based on the ranked order, including ranked features with one average importance greater than 0.5 (NDTI, GREEN, BLUE), ranked features with two average importance greater than 0.4 (NDTI, GREEN, BLUE, PGI, RED, SWIR1, NBR, NIR, SAVG), ranked features with three average importance greater than 0.3 (NDTI, GREEN, BLUE, PGI, RED, SWIR1, NBR, NIR, SAVG, SWIR2, CON, RPGI, EVI, PMLI, GRVI, MNDWI, DISS), and ranked features with 4 average importance greater than 0.2 (NDTI, GREEN, BLUE, PGI, RED, SWIR1, NBR, NIR, SAVG, SWIR2, CON, RPGI, EVI, PMLI, GRVI, MNDWI, DISS, CORR, VAR, VI, GNDVI, ASM, ENT, IDM, NDBI, NDVI).

We also evaluated the feature-by-feature iteratively F1-score curve based on the order of the ranked features. As shown in Figure 10, the trend of F1-score curves in 1995 (Landsat 5 TM), 2007 (Landsat 7 ETM+) and 2018 (Landsat 8 OIL) were similar, which reached around 0.8 before the top five features and then fluctuated slightly with the increase of features. Specifically, the F1-score reached the highest value (0.881) when it iterated to the top 22 features in 1995, the highest value (0.935) when it iterated to the top 23 features in 2007, and the highest value (0.940) when it iterated to the top 17 features in 2018. It verified that the method based on the multi-year average importance ranking of features could efficiently select those features that have an important impact on the classification accuracy.

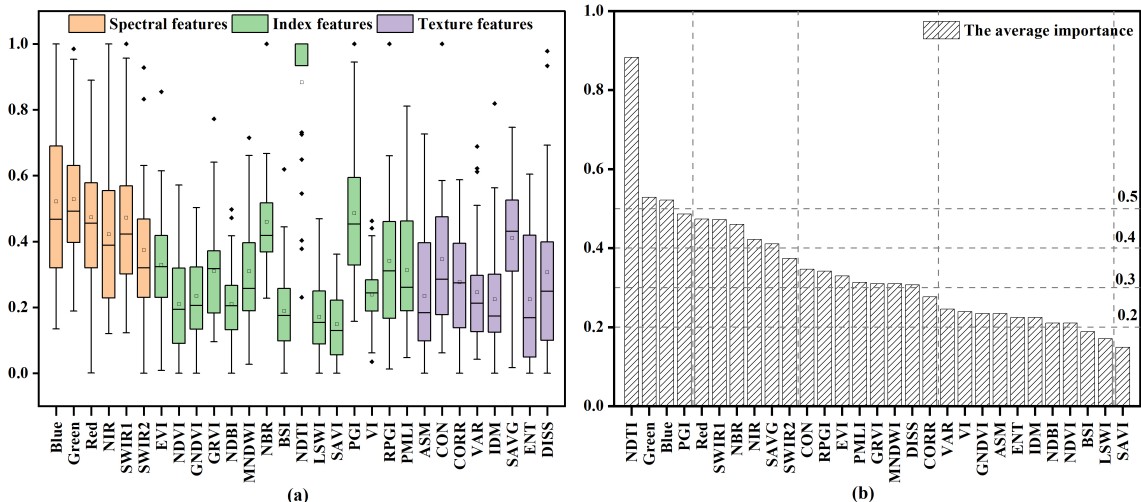

**Figure 9.** The importance of all features from 1989 to 2018: (**a**) the importance of each feature from 1989 to 2018; (**b**) the average importance of each feature over 30 years after ranking.

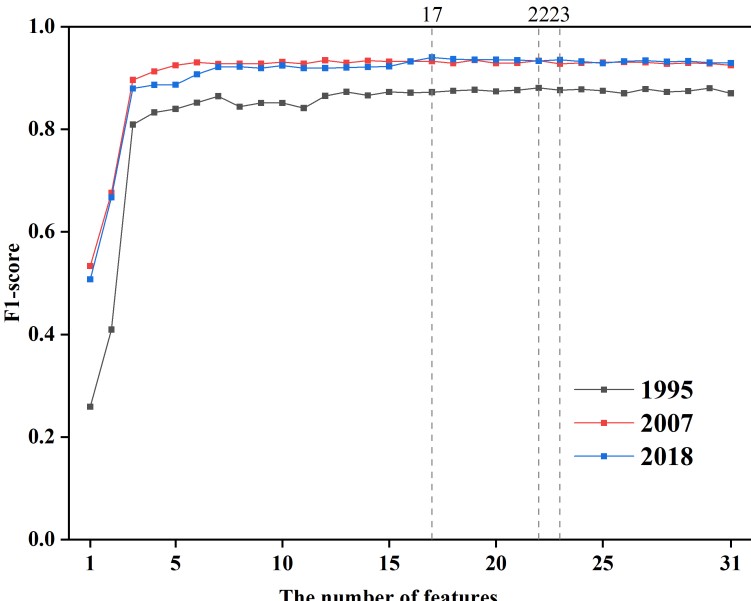

**Figure 10.** The change of F1-score curves of AG within the different selected years.

Therefore, a total of 10 sets of scenarios (Table 3) were designed to investigate the importance of different types of features for AG classification, and the average and standard deviation of F1-score in each scenario were obtained. From Figure 11 we can see that the average F1-score of scenarios 1 and 2 were equal and significantly higher than scenario 3, but the standard deviation of F1-score in scenario 2 was slightly lower than scenario 1, indicating that the spectral-based and index-based features were more effective in AG classification than the texture-based features, and the index-based features were more stable than the spectral-based features in multi-year AG classification. For scenarios 4 and 5 in which the index-based features and texture-based features were added to the spectral-based features respectively, their average F1-score were improved to 0.844 and 0.836 and the standard deviation of F1-score was reduced to 0.088 and 0.085, indicating that the combination of spectral-based features with index-based or texture-based features can both improve the average accuracy and stability over 30 years. Comparing the average and standard deviation of F1-score in scenarios 6, 7, 8, and 9, which were both ranked feature combinations based on the average importance of the features, we found that the highest average F1-score (0.869) and the lowest standard deviation of F1-score (0.063) were shown

in scenario 8, which was also the best performer among the 10 scenarios. All of these results indicated that although each feature showed certain importance for AG mapping, not all of them can achieve the highest and most stable classification accuracy. Therefore, we choose the results of scenario 8, in which the features with an average importance of more than 0.3, as the annual AG classification results.

**Table 3.** Different scenarios for feature optimization.

| Scenarios | Feature Combination (Count) | Scenarios | Feature Combination (Count) |
|---|---|---|---|
| 1 | Spectral-based features (6) | 6 | Ranked features 1 (3) |
| 2 | Index-based features (15) | 7 | Ranked features 2 (9) |
| 3 | Texture-based features (8) | 8 | Ranked features 3 (18) |
| 4 | Spectral and index-based features (21) | 9 | Ranked features 4 (28) |
| 5 | Spectral and texture-based features (14) | 10 | All features (31) |

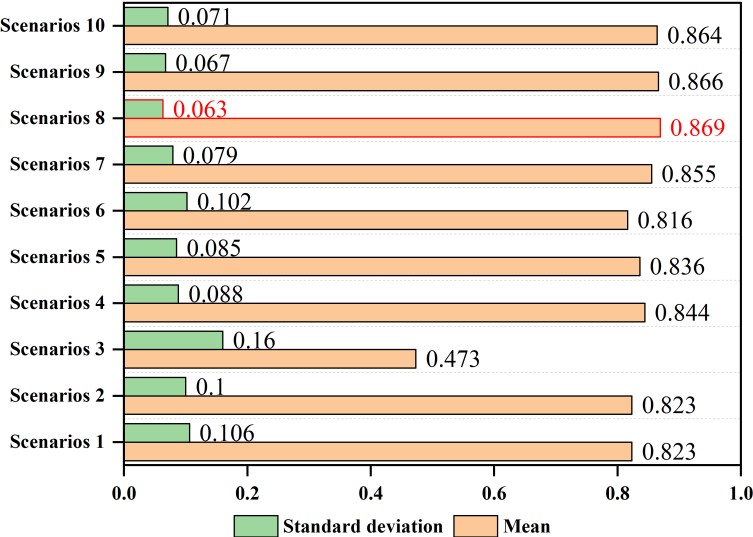

**Figure 11.** Comparison of accuracy of different feature scenarios.

### 4.2. Performance of the Temporal Consistency Correction

In this study, we designed 25 combinations based on the PTCC algorithm to explore the conditions under which the classification probability of adjacent pixels in the sliding window of each pixel needs to be corrected. As shown in Figure 12, when the threshold $a$ remains unchanged (i.e., the threshold for judging whether AG needs to be corrected to Non-AG), with the increase of threshold $b$ (i.e., the threshold for judging whether Non-AG needs to be corrected to AG), of the correction results gradually decreased and the standard deviation of F1-score gradually increased, indicating that for Non-AG with a large coverage area, it can be corrected directly if the classification results of the previous and subsequent years were AG (i.e., threshold $b$ equals 0.5). When the threshold $b$ remains unchanged, with the increase of the threshold $a$, the average F1-score of the correction results gradually increased and reached the maximum value (0.911), indicating that for AG with a small coverage area, the correction should be carried out when the classification probability of Non-AG in the previous and subsequent years were more than 0.8 (i.e., threshold $a$ equals 0.8). Considering the priority of the average F1-score, the correction threshold was finally determined as $a = 0.8$ and $b = 0.5$.

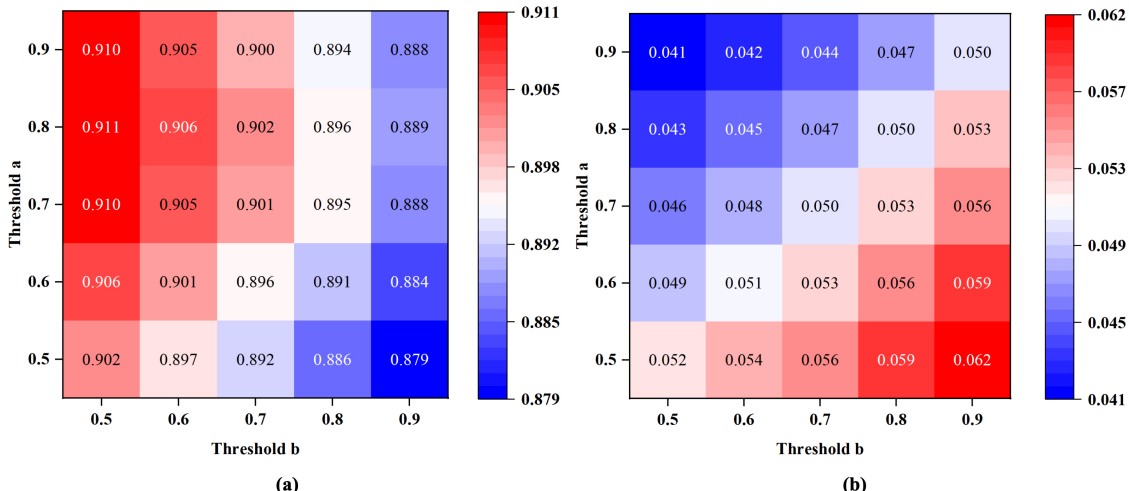

**Figure 12.** Comparison of 30-year average classification results after a temporal consistency correction under different classification probability combinations: (**a**) the average F1-score of the correction results, (**b**) the standard deviation of the F1-score of the correction results.

Comparing the F1-score of AG before and after correction year by year (Figure 13), it can be seen that the average F1-score of the corrected results has been effectively improved from 0.869 to 0.911. In terms of specific years, the most obvious improvement after the correction occurred in 1996 and 2003, with an improved F1-score by 0.190 and 0.101 respectively. In addition, for 2007, 2011, and 2012 using Landsat 7 ETM + strip problem images, the F1-score of the corrected AG classification was improved by 0.015, 0.028, and 0.047, respectively. In conclusion, the temporal consistency correction based on the PTCC algorithm effectively improved the quality of annual maps of AG in Shandong province, China from 1989 to 2018.

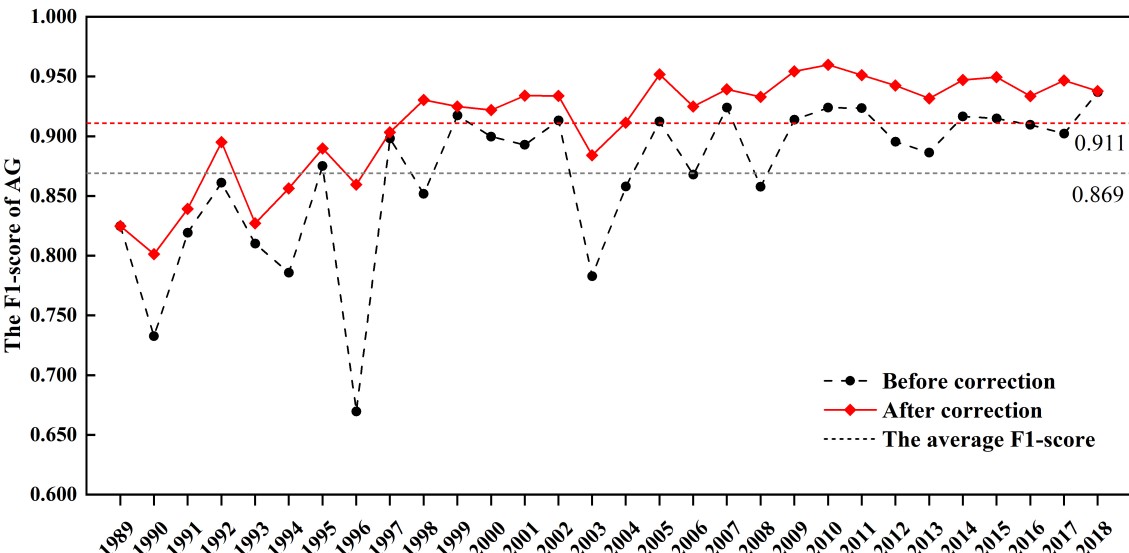

**Figure 13.** Comparison of F1-score of AG before and after temporal consistency correction year by year.

We further compared the results before and after implementing the temporal consistency correction by enlarging the details. As shown in Figure 14, the correction based on the proposed PTCC algorithm effectively eliminated the limitations caused by cloud cover or strip problems, et al. In region (a), the missed information underneath the cloud has been well restored for the cloud contaminated areas. Whereas for data missing areas that are due to the SLC-off issue of Landsat 7, it has been recovered through the temporal

consistency correction in the region (b). Besides, the underestimation of AG areas has been considerably complemented in the region (c).

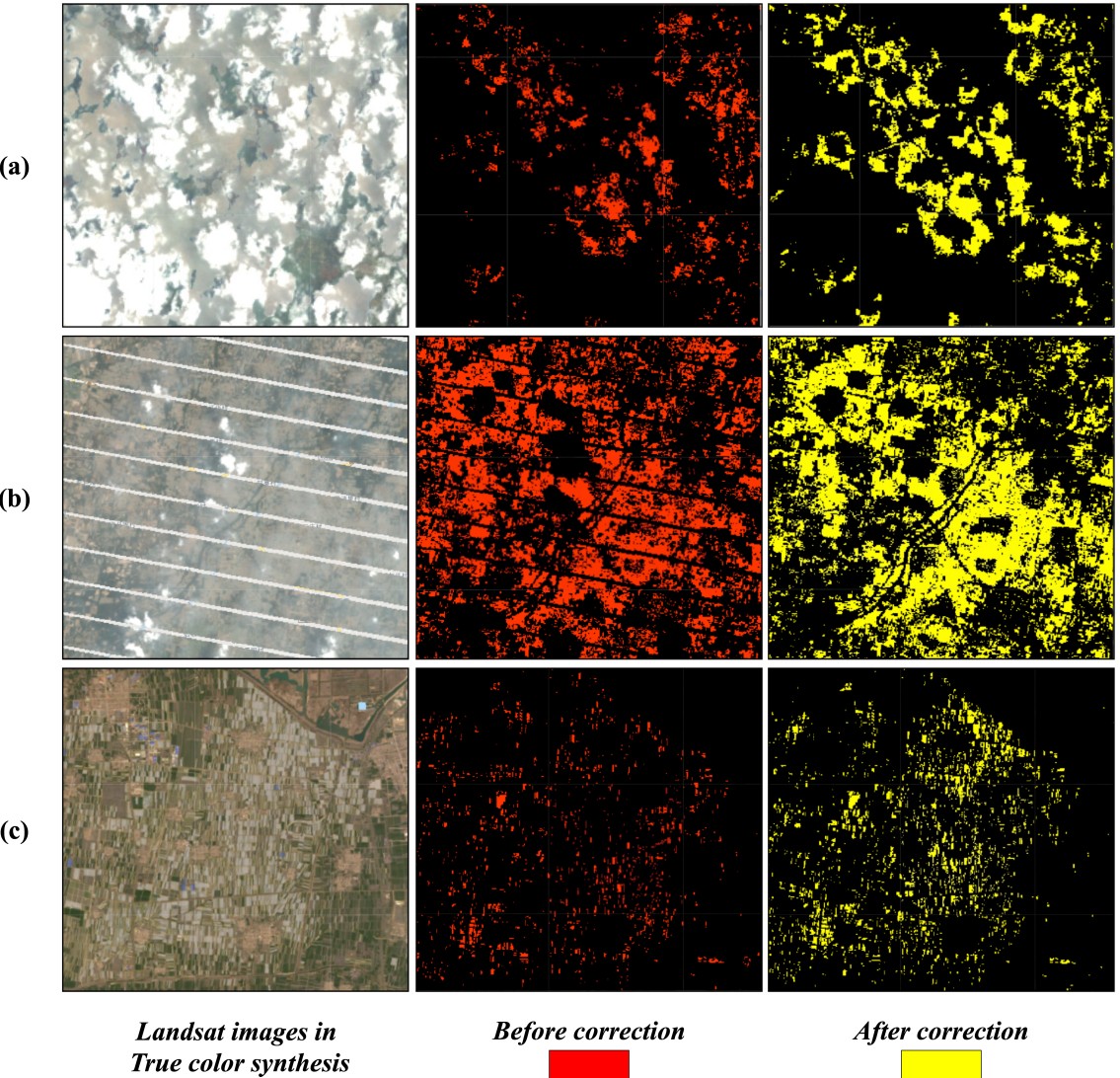

**Figure 14.** Improved classifications after implementing temporal consistency correction: (**a**) TM image acquired in 1996, (**b**) ETM image acquired in 2012; (**c**) OLI image acquired in 2018. Regions within ellipses are typical areas improved after implementing the proposed PTCC algorithm.

### 4.3. Annual Maps of AG in Shandong Province from 1989 to 2018

The accuracy of the annual AG mapping product was first assessed for each year via visually-interpreted test samples, respectively (Table 4). Overall, the accuracy of obtained AG sequences was stable and satisfactory, the F1-score of AG within all years are greater than 0.8, and for years 2005, 2009, 2010, and 2011, their accuracies are even better (above 0.95). From the classification accuracy of each year, the average UA of AG from 1989 to 2018 was 96.56%, the minimum value appeared in 2003 (93.67%), the maximum value appeared in 1991 (98.60%), and the standard deviation of UA in 30 years was 1.12%. From 1989 to 2018, the average PA of AG was 86.64%, the minimum value appeared in 1990 (68.31%), the maximum value appeared in 2005 (94.76%), and the standard deviation of user accuracy in 30 years was 7.60%. And the average F1-score of AG was 0.911, the minimum value appeared in 1990 (0.801), the maximum value appeared in 2010 (0.960), and the standard deviation of F1-score in 30 years was 4.33%. In summary, this product had a

good agreement between mapped pixels and referenced pixels, which can be used for subsequent analyses or for expansion modeling.

**Table 4.** Classification performance of AG in specific years.

| Class | Non-AG | AG | Total | PA(%) | UA(%) | F1-Score |
|---|---|---|---|---|---|---|
| Confusion matrix—1989 | | | | | | |
| Non-AG | 2429 | 6 | 2435 | 98.18 | 99.75 | 0.990 |
| AG | 45 | 120 | 165 | 95.24 | 72.73 | 0.825 |
| Total | 2474 | 126 | 2600 | | | |
| Confusion matrix—1999 | | | | | | |
| Non-AG | 2271 | 10 | 2281 | 98.48 | 99.56 | 0.990 |
| AG | 35 | 277 | 312 | 96.52 | 88.78 | 0.925 |
| Total | 2306 | 287 | 2593 | | | |
| Confusion matrix—2009 | | | | | | |
| Non-AG | 2053 | 16 | 2069 | 98.23 | 99.23 | 0.987 |
| AG | 37 | 555 | 592 | 97.20 | 93.75 | 0.954 |
| Total | 2090 | 571 | 2661 | | | |
| Confusion matrix—2018 | | | | | | |
| Non-AG | 1991 | 19 | 2010 | 96.79 | 99.05 | 0.979 |
| AG | 66 | 631 | 697 | 97.08 | 90.53 | 0.937 |
| Total | 2057 | 650 | 2707 | | | |

Figure 15 showed the mapped AG areas in each city of Shandong province in 1989, 1999, 2009 and 2018. The map series indicated that greenhouses in 1989 were mainly scattered in the northern plain areas (Dezhou, Jinan, Dongying, Binzhou and Weifang), and distributed in the northwest of Weifang (represented by Shouguang area) in 1999. By 2009, AG in Shandong province appeared in a provincial spread pattern, with concentrated distribution areas including Weifang, Linyi, Jining and Liaocheng, while further expanding on the previous basis in 2018. Generally, it showed the "C" shape spreading trend around the central mountainous area, especially in the north, south and west plain areas. In addition, Figure 16 showed the enlarged AG details in seven typical locations in 1989, 1994, 1999, 2004, 2009, 2014 and 2018, and we can see that the concentrated distribution of AG is often built according to the terrain, diffuses outward around the farmland around rural residential areas, and is divided by rivers and roads.

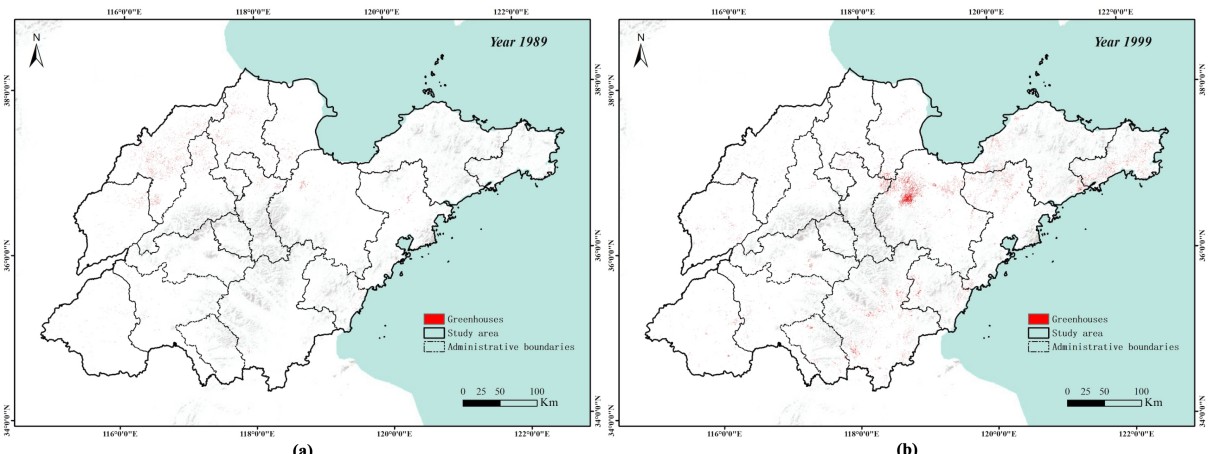

**Figure 15.** *Cont.*

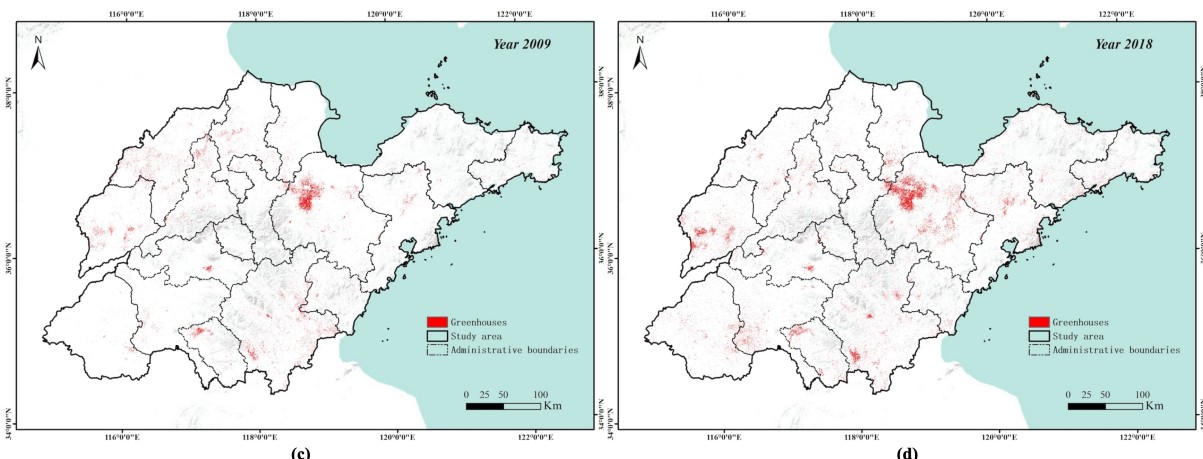

**Figure 15.** Spatial distribution of AG in Shandong province, China in 1989 (**a**), 1999 (**b**), 2009 (**c**) and 2018 (**d**).

**Figure 16.** Landsat image examples (ture color composite) for greenhouses in 1989 (**a**), 1994 (**b**), 1999 (**c**), 2004 (**d**), 2009 (**e**), 2014 (**f**) and 2018 (**g**).

## 5. Data Availability

The Landsat-derived annual AG dataset herein is now available in the public domain at https://doi.org/10.5281/zenodo.5633605 (accessed on 15 November 2021). This dataset was tagged in GeoTIFF file format, with a spatial resolution of 30 m under the WGS84 geographic coordinate system, named as "AG_SD_CHN_v01.zip". A detailed description of the classification system is also provided. The uploaded imagery can be processed using GIS software such as ArcGIS or QGIS.

## 6. Future Works

Future works should consider several aspects: first, in order to achieve high spatial accuracy and temporal consistency of the final annual AG mapping products over a long time at the provincial scale, the sample acquisition process was still in accordance with the sampling method year by year. The idea of transfer learning should be introduced to explore the spatial and temporal generalizability of the classified samples, reduce the labor cost and improve the intelligence of classification under such scenario. Second, limited by the time span of the study, we can only use Landsat-derived images with a spatial resolution of 30 m, with the development of the Earth observation satellite system and the accumulation of high-resolution remote sensing data, images with higher spatial resolution should be further used in future studies to achieve fine classification of different greenhouse types. Third, the mapping product generated here have a wider coverage and higher spatiotemporal resolution than statistical data and other similar studies. As one of the most typical provinces for the development of protected agriculture in China, Shandong Province has received attention in the fields of industrial development, economic geography and environmental science regarding the socio-economic and ecological impacts brought about by the development of greenhouse-related technology, and it is also a direction worth exploring for future research to break through the disciplinary barriers to achieve interdisciplinary sharing and analysis of this dataset.

## 7. Conclusions

With the rapid urbanization and agricultural modernization in China, a sequential and fine-resolution AG product is the fundamental information to implement informed and sustainable protected agriculture management. However, previous efforts to map AG have generally used visual interpretation and supervised, and unsupervised methods, and mono-temporal or multi-temporal images at regional scales, while fine-resolution annual AG maps at large scales have rarely been investigated in the current literature. Therefore, to better understand the expansion of AG in the representative province of China, we generated the first Landsat-derived annual AG dataset from 1989–2018 in Shandong province based on the GEE platform. Combining vegetation growth with the phenological information, we found that DOY 73-136 (from 15 March to 15 May) is the best mapping window within a year, which is critical in distinguish AG, cropland and plastic-mulched land in the study area. Furthermore, by employing feature optimization via calculating the importance of each tested feature, the mapping accuracy of each year has been considerably increased. In addition, the proposed temporal consistency correction algorithm played a crucial role in the eventually obtained products, with an improvement of approximately 5% on average. It is strongly recommended that temporal consistency should be corrected based on classification probability when mapping AG over multiple times. Eventually, we got accurate and long-term AG dynamic maps in the most representative province of China, which have an average UA of 96.56%, an average PA of 86.64% and an average F1 accuracy of 0.911. Therefore, the AG maps in this study could provide important reference and scientific guidance for the efficiently planning and managing of protected agriculture. Combined with other data (e.g., environmental data or socio-economic data), it also could serve as the higher spatiotemporal resolution dataset for more comprehensive characterization of its ecological and economic impacts.

**Author Contributions:** Conceptualization, C.O.; Investigation, J.Y.; Methodology, C.O. and Z.D.; Software, T.Z. and B.N.; Validation, Y.L.; Writing—original draft preparation, C.O.; writing—review and editing, Q.F. and D.Z. All authors have read and agreed to the published version of the manuscript.

**Funding:** The research was funded by Ministry of land and resources industry public welfare projects (No: 201511010-06).

**Institutional Review Board Statement:** Not applicable.

**Informed Consent Statement:** Not applicable.

**Data Availability Statement:** All relevant data are within the manuscript. The AG dataset produced in this study is now freely available at https://doi.org/10.5281/zenodo.5633605.

**Acknowledgments:** The authors would like to thank the google earth engine (GEE) team and the user community for their useful feedback during this research process. And thank the journal's editors and anonymous reviewers for their kind comments and valuable suggestions to improve the quality of this paper.

**Conflicts of Interest:** The authors declare no conflict of interest.

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
