# Peer review of "Landsat-Derived Annual Maps of Agricultural Greenhouse in Shandong Province, China from 1989 to 2018"

_remotesensing, doi:10.3390/rs13234830_

Round 1
Reviewer 1 Report
This study was aimed at mapping the annual agricultural greenhouse (AG) from 1989 to 2018 in Shandong province, China. For this purpose, authors have used multi-period Landsat dataset processed on Google Earth Engine platform, and an annual remote sensing mapping method of AG oriented to the provincial area and long-term period was proposed by integrating the remote sensing characteristics analysis, mapping window selection, classification feature optimization and temporal consistency correction for annual classification results.
In general, the dataset produced in this study have important implications for sustainable agricultural development and fill a relevant research gap. The research idea is clear and well designed, and the manuscript is well written with adequate description for each manuscript section and easy to read though needs very few revisions. A few points are listed below:
- L36, need to cite the source of statistical data like L32;
- L49, suggest to point out the linkage with China's cropland protect policy;
- L82, why “the AG is easy to be mixed with bare cropland and plastic-mulched land at provincial scale”? what about other scale?
- “Provincial area” in manuscript need to revise to “large-scale area”, because the scope of the provincial level varies greatly from country to country;
- L113, “15.775%” suggest to revise to “15.78%”;
- Section 2.2, why used the SR level Landsat data?
- L150, Google earth images used may be subject to copyright issues and should be cited accordingly;
- Figure 5, “greenhouses” or “greenhouse”? Needs to be unified throughout the manuscript;
- Section 3.6, why is the overall accuracy not used?
- Figure 10 (b) is not part of this paragraph and should be placed after Table 3;
- Figure 15, same issue as Figure 5;
- L495, an extra punctuation mark.
Author Response
Dear Reviewer,
Thanks very much for taking your time to review this manuscript. we really appreciate all your comments and suggestions! Please find our itemized responses in below and our revisions in the re-submitted files.

Reviewer 2 Report
This manuscript, entitled as “Landsat-derived Annual Maps of Agricultural Greenhouse in Shandong Province, China from 1989 to 2018”, used Landsat images to map and explain the agricultural greenhouse changes. The classification was oriented to a sequential period and large-scale. Authors also proposed a corresponding classification framework that integrated remote sensing characteristics analysis, mapping window selection, classification feature optimization and temporal consistency correction. The results are statistically sound, and the authors performed a detailed discussion on different mapping window, test features as well as classification probability. Overall, the manuscript is well-organized.
The topic of this paper clear matches with the scope of Remote Sensing and it could be an interesting contribution for the readers (in particular, the dataset made public in the paper). Thus, it has the potential for publication after revision.
- Line 29, what does a balanced food supply mean exactly? In terms of time, space or food types?
- The first and second paragraphs of the introduction are proposed to be modified and combined.
- Line 61-63, RPGI should also be added in, as you described in Section 3.3.
- Line 82, “provincial scale”, such an expression is prone to misunderstanding, and it seems to be on a “large scale”.
- Line 113, “15.775%” the number format should be unified.
- Section 2.3, is the number of samples in each sampling grid the same?
- Table 1, why is there such a big difference in the number of AG and non-AG? and how does an unbalanced sample affect the classification results?
- Line 175-176, why “…but theoretically, its seasonal variation is more stable than that of cropland or forest.”? Needs more explanation.
- Figure 5, “Agricultural greenhouses” needs to revise to “Agricultural greenhouse”, and the same revision should be applied to the rest.
- Line 236, what’s the role of image composition within the selected mapping window?
- Line 284, the ID of RF in GEE is not required to be presented in academic papers.
- PTCC, why is only the correction of the time dimension considered, but not the spatial dimension?
- Section 3.6, why not consider using overall accuracy and Kappa coefficient?
- Line 354, “Landsat5/7/8” missing a space.
- Figure 10, F1-score should be revised to the F1-score of AG.
- Line 381, scheme or scenario? should be unified.
- Figure 10 (b) needs to place after table 3.
- The "C" shape, as you described in line 438, could be present in Figure 14 (d).
- Line 455, “In future research” needs to delete, due to repetitive expression.
- Line 495, “…and economic impacts.”, an extra end.
Author Response

(The authors gave the same response as above.)

Reviewer 3 Report
The authors study an important topic related to landscape management.
The introduction clearly explains the state of the art and the purpose of the work.
The work was carried out with in-depth bibliographic analysis and in a good scientific way.
It appears that the researcher team is fully integrated.
Author Response
Dear Reviewer,
Thanks very much for taking the time to review this manuscript.